# A Study on the Xiuxing of Contemporary Horchin Mongolian Shamanism

**Yumin Lun * and Xiaomei Dong**

Social Science Research Center, Zhejiang University of Science and Technology, 318 Liuhe Road, Xihu District, Hangzhou 310023, Zhejiang, China; dongdxm0319@163.com

*   Correspondence: lunym27@163.com

**Abstract:** Research has been carried out on the procedures for recruiting and training shamans among the Horchin (mainly in Tongliao City, China). This well-known problem is crucial to the development of Horchin shamanism. If a potential shaman wants to complete the transition from an ordinary person to a shaman, they need to repeat religious practices, progress spiritually, learn, and deal well with the role between their daily life and religious life. This process of Xiuxing is full of hardship. However, the issues surrounding the requirements, influencing factors, and evaluation criteria has received little attention. We have been conducting fieldwork in the Horchin area since 2013, have continuously tracked and interviewed more than 100 shamans and prospective shamans, and have obtained much fieldwork data. Through the collation, induction, and comparative study of these materials, we found that Horchin shamans are required to study the knowledge and skills of shamanism, respect their teacher, obey their principles, fulfill the duties and obligations of a shaman, and devote their lives to serving the local community. We also found that Horchin shamans are struggling to adapt their religious practices to the belief systems of the contemporary Chinese world. We also found that it is believed that, in the region, a successful shamanic career presupposes not only knowledge of rituals but also compassionate and principled behavior with respect to the clients and the community.

**Keywords:** Horchin Mongolian; shamanism; Xiuxing; inheritance of teachers' spirit; duty and obligation; human nature

## 1. Introduction

On 19 September 2013, the Third Forum on Religious Anthropology was held at Lanzhou University. Chinese scholars Yang Derui 杨德睿, Chen Jinuo 陈进国, and Huang Jianbo 黄剑波 suggested that we pay greater attention to people's "inner worlds", meaning people's internal understanding of religion. Anthropology has traditionally focused on the externally observable, the structures and functions of various social systems—which, in this case, are religious. A greater focus on the internal and intangible, the scholars said, could lead to a better understanding of people's experience and motivations (Chen 2017). With their efforts the phrase "Xiuxing[1] anthropology" was coined.

According to the Chinese dictionary 汉语大词典, the word "Xiuxing" has three meanings: (a) to leave home to learn Buddhism or Taoism, (b) to cultivate virtue, and (c) to possess good character (Chinese Dictionary Editing Committee 汉语大词典编辑委员会 1986, p. 1372). The word is often used with religious connotations. There are certain similarities between religious Xiuxing and the

---

[1]    This is a Chinese local word pronounced like syo-sying or sho-shing.

spiritual practice of Western Christianity. Both traditions maintain that followers should use a set of core doctrines and canons to guide their beliefs and regulate their behaviors (Liang 2017, p. 25). However, many psychologists, represented by William James (1902, p. 355), prefer to divide religion into two aspects: institutional and individual. They tend to place a greater emphasis on the individual aspect. For example, James believes that the prayer of a Christian is a kind of internal communication or dialogue with sacred power.

Similar to the views of Western scholars, Liang (2017, p. 25) argues that Xiuxing is a series of self-restraint and transformation in the aspects of concept, experience, and behavior that individuals or groups take the initiative to achieve transcendence and holiness. More Chinese scholars believe that religious Xiuxing is related not only to individualized behavior but also to groups and to group behavior. Yang Derui 杨德睿 (Yang et al. 2017, p. 123) argues that, for religious believers, Xiuxing is "a variety of learning activities that particular religions' followers take to become more ideal religious believers—superbly sacred," or "to study how people gradually learn and embody a certain type of religious sentiment, feeling, cognition, will, imagination, practice, etc. which is cultivated by religion." Yang Derui and Huang Jianbo 黄剑波 (Yang and Huang 2017, p. 13) also argue that religious Xiuxing is a process in which people are transformed from the status of outsiders into that of believers through the gradual strengthening of their knowledge, beliefs, and enthusiasm for the religion. Chen Jinguo 陈进国 (Chen 2016, p. 46) holds that the connotation of Xiuxing involves a unity of knowledge and action, including the superior pursuit of cultural order or rationality, such as ritual behavior or taboos (Di 涤, Jing 儆, Jie 洁), sacred understanding (shame), greater knowledge (experience or transcendence), and moral or spiritual cultivation.

According to the view of these advocates, we can define the word Xiuxing as daily, self-cultivation of certain practices and moral subjectivities, fulfilling the requirements of Xiuxing requires people to pay attention not only to the transcendence of the religious world but also to the transcendence of daily life. We also believe that religious Xiuxing is not only a personal matter of religious believers but is also closely related to groups and society. Any religion requires appropriate religious models. These models are appropriately followed by religious believers to improve their religious cultivation, their moral perception, and their religious practice. Any religion requires doctrines, religious precepts, and taboos. These elements not only provide effective constraints and guidance for the behavior of religious believers, but also promote a type of awakening of virtue and spirituality, which in turn all contribute to the maintenance of the physical and mental well-being of individuals or social groups.

We have long been interested in Mongolian shamanism, especially from the Horchin Region, particularly its preservation and its practice in daily life. Inspired by the concept of Xiuxing anthropology, we wanted to explore the various elements of Mongolian shamanism from the perspective of the shamans themselves. Horchin shamanism does not have formally codified religious doctrines like those of the major world religions, nor does it have a systematic set of guidelines for interpreting those doctrines. However, there are well-established requirements for a shaman to demonstrate spiritual and religion skill progress recognized by the shaman group and local people through Xiuxing (or religious practice).

Since August 2013, we have been conducting fieldwork on the shamans in the Horchin region. We interview with more than 100 shamans and prospective shamans around the core issues of the reason you become a shaman, the initiatory process of an ordinary person into a shaman, and the shamans' growth and improvement. Among these people, more than 60% person can speak Mandarin, and most of those who cannot speak Mandarin but are only Mongolian are elderly. The goal of Horchin shamanism is to become an ideal shaman as required by the Horchin social evaluation. This requires the shaman to engage in Xiuxing persistently in his daily life.

From our fieldwork, we find that the transmission of Horchin shamanism relies heavily on a mentoring system involving a *shifu* 师父[2] who teaches and apprentices who learn. It would be difficult for ordinary people to learn ritual skills and to advance morally and fulfill the shaman's duties and obligations on their own. The smooth progress of Xiuxing depends on the *shifu*'s instructions and on the apprentice's docility. It also relies on the character and personality of the *shifu*, as well as his level of knowledge.

## 2. The Xiuxing Way to Become A Shaman

In the Horchin Region of Inner Mongolia, before becoming a shaman, ordinary people who possess shamanic potential usually have one or another type of "shaman disease"[3] caused by the souls of their own ancestors who were shamans before, which was called *sitügen* 希图根[4] by the locals. They also need to have experienced some unusual events. According to our fieldwork, the most common symptoms of shaman disease include sudden madness, mental disorder, mumbling, dreaminess, and other abnormal mental phenomena. There may also be abnormal physical symptoms, such as feeling too weak to get out of bed, pain in the legs and feet, and other difficulties in moving. Unusual events include unexpected catastrophes such as sudden traffic accidents but with no bodily pain, slips and falls, falls from great heights with no bodily hurt, and a loss of livestock. Since ordinary hospitals do not handle these types of phenomena, patients may turn to a shaman.

Our fieldwork suggests that the process by which a future shaman looks for a *shifu* teacher may still be little understood by mainstream academia. The persons we interviewed in Horchin indicated that future shamans meet their *shifu* through what they perceive to be predestined connections and mysterious guides. For example, the prospective shaman is guided in his or her dreams (usually by his own *sitügen*) to a certain place where they will find their future *shifu*. Interestingly, the *shifu* may also know that a destined apprentice will come soon and may even wait for the apprentice at home. In their own words, their *sitügen* spirit guide was involved ahead of time, much in the nature of a mentor, good friend, or peer. It may also be the case that the *sitügen* of the *shifu* was deeply admired by the *sitügen* of the future shaman. The *shifu* judges the apprentice's *sitügen* through special shamanic skills entailing visual observation or smell, or through religious rites such as "requesting a spirit for the body."

The *shifu* then applies certain skills, such as shamanic songs and rituals, to treat apprentices according to established procedures. The beginning of the treatment is also the beginning of learning shamanic traditions, of training in shamanic skills, and of guiding the future shaman through to the enlightened state of Xiuxing. The first important skill for a *shifu* to transmit to his apprentice is the skill of inviting his *sitügen* to possess his body, which is generally done by entering a trance state (*kaikou* 附体状态开口说话)[5]. For example, on 1 October 2014, the shaman Don Huzler唐呼日勒 of Yolimoden Town 腰林毛都镇 held a ritual called *kaikou* for his disciples at home. The ritual went as follows:

---

[2]    The Chinese respect for the teacher. In Horchin Shamanism, there is a relationship between teaching and learning between the shamans and their disciples. The disciples generally call their own shaman teacher as "shifu."

[3]    The term *shamanism*, as it is used here, does not carry the late nineteenth- and early twentieth-century views of shamanism as "hysterical, neurotic, epileptic, schizophrenic." It refers to the Inner Mongolian people (*külög*) in the Horchin Region who have control over the "*sitügen*" spirits. According to our fieldwork, before they know that they may be potential shamans, most of them may experience mental and physical problems or accidental events that suggest a shaman's disease.

[4]    *Sitügen* is a local word that used only in shaman group. Generally, *sitügen* has several types: (a) Some are the souls of the shamans' own ancestors who were once shamans. After he or she died, their soul would find a person from his or her descendants. Most of them would find descendants with the same gender. This type of *sitügen* is their main deity, and the shamans often through a special ceremony to request their *sitügen* leave away from the place of residence by the possession to help them handle some things. (b) Some *sitügen* would find a person without any blood relationship. (c) Sometimes the animal spirits of the shamans, such as fox, white snake, and mouse, are called *sitügen*. Most of the time the local shamans use this word to refer to their own personality spirit.

[5]    *Kaikou* 开口 is the phenomenon where an individual goes into a trance as a spirit (his or her *sitügen*) possesses his or her body and opens the mouth to speak through it. *Kaikou* is highly significant, because it marks the healing of a client and the entry of a trainee into the ranks of shamanism.

the apprentice stood in the middle of the room with his hands folded and his eyes closed, facing the door of the west room (which opened to the east gate). Several of his peers stood around him. They played drums, sang songs in praise or devotion to their own spirits, and, under the guidance of the *shifu*, took protective measures to prevent accidents that could be caused by unskilled movements while inviting the disciples' *sitügen*.[6] Some of the apprentices there had already spoken words during a trance state, known as "*sitügen* possession," while others were still learning. Newly formed shamans stood by to watch and give moral or spiritual support to those still undergoing training. After one apprentice succeeded in the state of possession, another would continue until all those who wanted to invite in their personal spirit had done so. This process is not only a course of treatment but also a learning process for the future shaman. In addition to learning how to invite the spirit into the body, playing musical instruments, and singing with the teachers, the apprentices have to participate in sacrificial activities hosted by the *shifu*, such as animal sacrifices to the *oboo* 敖包and to other spirits.

If the person affected with such a "shaman illness" eventually wants to become a *shifu* and use such complicated, orally transmitted religious knowledge, they have to follow the *shifu* for a long time and strive for as many opportunities as possible to participate in the *shifu*'s activities. Some rituals are challenging even to the *shifu*, and if the future shaman misses these learning opportunities, they will never gain reliable knowledge or skills. Many contemporary shaman trainees therefore spend considerable time following their *shifu* while they are still alive. Some of the older shamans mentioned in this article have had the experience of following their *shifu* for years on end, such as Qian Yulian 钱玉兰, Don Huzler 唐呼日勒, and Bai Mochuoahi 白毛绰海. The bonds of affection that emerge between the *shifu* and the apprentices may become quite strong, and in the process the apprentices develop a strong desire to continue learning with the *shifu*. The *shifu* will focus on instilling perseverance, understanding, and good character in their apprentices, in addition to leading them to participate in various shamanistic activities in order to accumulate practical experience.

A future shaman will also face considerable financial pressure. The obligation for apprentices to follow their *shifu* year after year affects their family life and their income. Some shaman trainees come from economically comfortable home situations or live in close proximity to the *shifu*'s home. This allows them to follow and learn from the *shifu* over an extended period of time. Some poorer trainees may actually live in the *shifu*'s home while holding regular jobs to make ends meet. These outside economic activities not only prolong their *kaikou* time but also hinder their ability to systematically and comprehensively absorb and learn the requisite religious knowledge and skills.

For one informant whom we interviewed, this long process of education and spiritual development of quasi-shaman lasted for nearly ten years. From 2005 to 2015, the future shaman Wu Bo (pseudonym for an unmarried male farmer) started his *kaikou* with the help of his *shifu* shaman Don and was gradually cured of the shaman disease symptoms. There are also some economic pressures due to the cost of animal sacrifices. Animal sacrifices are an important aspect of the rituals of shamanism and play a significant role in the cosmological calendar. For example, on July 7 and September 9 of the lunar calendar, the shaman will slaughter a pure-colored fat castrated ram in tribute to local deities. The purchase of these animals puts a significant economic burden on the shaman but is considered an obligation that the shaman must perform. Failure to make these sacrifices could result in punishment by an offended deity, thus putting even greater pressure on the poorer shamanic trainees.

---

6　　After the disciple under the trance state sat down on the bed, *shifu* Don asked to him, "Please tell me your name? What is your position at *bayiri* (the place of the Horchin shamans sitügens' residence) and *suur* (the last time the *sitügen* as a shaman's residence)?"(the meaning of the two words *bayiri* and *suur*, see Zhao 2015) He couldn't open his mouth, but we felt that he was struggling hard to open it. Don pried open his clenched teeth with the tip of a dull sword and pressed down on his tongue. Finally, he spoke a few words from his teeth: Alatan (his *sitügen*'s name). When the people present heard it, some made a sigh of exclamation. His wife was very excited after hearing it, and she could not restrain her excitement. After the end of the possession state, we asked him how he felt. He said that he had been expecting this day for a long time and that he now feels very comfortable both in his body and heart. His wife said that this day was even more important than the Chinese New Year. Don said, "well, finally he opened his mouth up" (*Kaikou*). The other disciples came to congratulate him and his wife and said that *Kaikou* was a good thing.

When the apprentices are ready to enter a state of trance (*kaikou*) under the guidance of the *shifu*, the *shifu* usually asks them to prepare some basic shamanic instruments, including clothes, caps, and bronze mirror belts. The *shifu* allows the apprentices to prepare the instruments before the *kaikou*. They then continue to invite the spirit every day until the apprentice learns how to routinely speak to others in a state of *sitügen* possession. At this stage, there are several options for new shamans. Some people take the mantle and adopt shamanism as their career. After recovering from their illness, they continue to learn and practice with their *shifu*[7]. Others may have been simply intent on achieving a cure for their diseases, with no interest in further engagement in the practices of shamanism. However, it is generally believed that, if they do not practice their calling, then their shaman diseases are likely to recur at some point in the future, possibly even years later. At that time, they will need to go back to their *shifu* and once again deal with their *sitügen*.

There are many similar cases in the countryside. In fact, it is considered a common occurrence in the rural areas of Horchin. One example is a female shaman named Bao Wuyun 包乌云, who raised more than 100 sheep to augment her family's income. The long hours she put in with the sheep made it difficult to practice her shamanic training. After five years (in 2015), she fell ill again with the symptoms of reduced energy and mobility associated with shaman diseases. In addition, her flock of sheep fell ill and all died one after another. One day, accompanied by her family, she once again turned to her *shifu* Wang Nuna 王努娜 for help. Her *shifu* criticized her for angering the *sitügen* by not fulfilling her obligations. She advised her to plead with her spirit to relent from the punishments. In the evening, with the help of her *shifu*, the female shaman successfully summoned her *sitügen* to communicate with the harmful spirits who had entered and weakened her body. At that moment, her left hand started to slap her face strongly, stopping only after the *shifu*'s comforting words. The *sitügen* who was possessing the offending shaman's body was showing his anger and was criticizing the *külög* 胡鲁格 (female shaman) harshly. To appease the spirit, the *shifu* entered into a contract with the *sitügen*, promising that the negligent *külög* would change her behavior and would in the future frequently invite him to possess her and to supply offerings on time. The goal was to move the offended *sitügen* to forgive the negligence of his *külög* and to begin protecting her once again. It should be pointed out that the people who request the help of a shaman do not all want to become shamans themselves. They invite a spirit into their home at the request of the *shifu* and are unharmed by the experience.

## 3. The Xiuxing Way as A Shaman

Strictly speaking, the new shaman's ability to clearly communicate and talk to others after being possessed by their *sitügen* is only a basic qualification, achieved at the initial stage of the shaman's growth. At this phase, the shaman acquires only some basic and fragmentary shamanic knowledge. They are still not competent enough to hold various sacrificial ceremonies, such as sacrifice natural ritual and national ritual, and they cannot do the ritual to remove persecution from ghosts and evil animal spirits. Moreover, they cannot heal others in the shamanism way, but healing is a major role for Horchin shamans. They must continue deepening their knowledge and skills in Horchin shamanism. Therefore, a continual learning process is central to the religious Xiuxing of inductees, from the patient stage up through and beyond the shaman stage. In our fieldwork, we often heard many older shamans, such as Se Renqin (1925–2007), a very famous shaman, say to their apprentices: "Don't think that you are great when you can invite a spirit to cure an illness. There is still a lot of knowledge about *Boo*, so study hard"[8]. Through his oral maters we know that, in traditional Inner Mongolian society (before 1949), apprentices not only have to learn from their own *shifu* but must also go out to learn from other

---

[7]    There are both professional shamans and part-time shamans. The latter do not make a living from shamanic activities, but perform ritual activities when someone asks for their help.

[8]    Recounted by Zhao Furong, recorded by Lun Yumin.

famous shamans. Today the apprentices' learning requirements are the same. Moreover, potential sources of knowledge are not limited to the shaman group. Scholars, cultural workers, and journalists are also approached by shamans requesting different types of guidance.

The most important condition for Xiuxing is for the apprentice to learn with the *shifu*. The shamanistic knowledge system is very complex. Having no written doctrines, it must be transmitted orally by the *shifu*. The process of education for a young shaman is a mix of oral transmission of knowledge imparted by the *shifu* in everyday interactions. However, some knowledge can be learned only by taking part in ceremonies or rituals performed to treat a specific sickness or person. A core belief is respect for the spirits; they should summon the spirits only when there is a need. If an apprentice cannot follow the *shifu* on a regular basis over an extended period of time, he or she may miss some key ceremonial activities that may be requested by the public, and they will not be able to learn from his *shifu* the skills required to handle such situations. If new shamans are arrogant or excessively self-interested, if they lack humility or perseverance, they will at most learn only some fragmented techniques of shamanism, and the ritual activities they carry out will be largely superficial. This will cause them to fail in the competition with other shamans over time, in addition to hindering their path toward spiritual progress.

In addition to learning from and with the *shifu*, the shaman should also actively learn the relevant cultural knowledge and strive to improve his cultural literacy. Historically, Mongolian shamanism has seen periods of growth and decline. Inner Mongolian shamanism flourished for thousands of years and was once even the "state religion." With the spread of Tibetan Buddhism in Mongolia and the suppression of shamanism by various rulers, Mongolians gradually converted to Tibetan Buddhism (Lun et al. 2010), and the influence of shamanism was gradually reduced to a region of Inner Mongolia, the Horchin Region. Modern Horchin shamanism still struggles to stay afloat, with the popularity of shamanic religious practices continuing to wane (Bai 1986). During the Cultural Revolution, shamanism was portrayed as one of the four "great evils" and was targeted for criticism and eradication. Many shamans were forced to stop performing the boo dance. Many religious artifacts were burned or destroyed, and Horchin shamanism was pushed nearly to the point of extinction. Fortunately, through the joint efforts of scholars, government officials, and shamans themselves, Horchin shamanic culture has become better preserved. Myths, legends, and traditional dances have been recorded, while relics and artifacts have been collected in museums, allowing for research into Horchin shaman heritage. In fact, many of the local scholars and cultural workers who participated in the original shaman research interviews are still alive and can thus provide guidance on various areas of shamanism.

For example, a shaman named Qian Yulan 钱玉兰 wanted to go to Beijing to perform a Horchin shaman dance. However, she was not familiar with the dance. Therefore, she consulted a scholar of shamanism named Bai Cuiying 白翠英, who worked in the Tongliao City Art Research Institute 通辽艺术研究所 and who had interviewed her *shifu* Se Renqin 色仁钦. Under Bai's guidance, Qian was able to perform the dance successfully, in addition to learning many other shamanic traditions. These traditions include festivals, cultural taboos, and other aspects of Mongolian culture. Within Inner Mongolia, the Horchin Region in particular is a melting pot of Han Chinese and minority Hui, Manchu, Tibetan, and Mongolian cultures.

The economy of this region has been changing from agrarian to industrial. Consequently, the Mongolian cultural atmosphere is no longer as strong as it is in the more pastoral regions such as Hailar 海拉尔[9]. People are becoming less knowledgeable about ethnic minority cultures. Shamans must therefore spend more time deliberately studying their culture. Highly educated shamans, or those with high intellectual ability and a strong desire to learn, often have long and prestigious careers.

---

[9]　For example, although people in the Horchin Region also speak and write the Mongolian language, many no longer wear Mongolian clothing. They refer to themselves as "short-robed Mongolia" 短袍蒙古. They say, "Although we are Mongolian, many customs are already like the Han, but we are not Han." (我们蒙古人不是蒙古人，汉人不是汉人.)

In addition to learning with a *shifu*, a shaman should take the initiative to improve his own practice through self-learning. Communication skills are especially important. A young shaman Hai Qing 海青 was adept at *kaikou* and was even able to invite the spirits of birds and tigers. However, he spoke with a stutter, which hindered his confidence and made him less able to communicate smoothly. Because of this lack of confidence, he would often refer potential patients to his *shifu*. This prevented him from gaining enough experience to advance to the next stage of Xiuxing. This demonstrates the importance of communication skills in the process of learning to be a shaman.

There are many strands of shamanism, each with its own songs, rituals, musical instruments, and approaches to problem-solving. People-to-people exchanges are therefore necessary in order to deepen understanding of the differences among the different strands, thus facilitating the preservation of shaman culture. Thanks to modern transportation and communication technologies, as well as to the help of institutions promoting the preservation of Cultural Heritage, shamans in the Horchin Region are finding it easier to maintain relatively closer communications with each other. Some *shifus* have even restored the tradition of shaman competitions[10].

## 4. The Religion Moral (or *Tao*) Xiuxing as A Shaman

In addition to learning the actual knowledge and skills (术, *shu*) of the craft, new shamans also need to strengthen their mind, character, and self-discipline (道, *tao*). Career shamans cultivate a high moral discipline and observance of the religious commandments, earning them the trust of the community and enabling them to act as role models for shaman apprentices.

Horchin shamanism does not have a unified code of conduct, but it does have practices that are generally observed by the shamans. For instance, shamans must respect their *shifu* and the spirits, they must maintain strict discipline and avoid religious taboos, they are forbidden to use any kind of witchcraft or black magic to cause anyone harm, and they may not promote false interpretations of shamanism. These principles are crucial for achieving Xiuxing, and they are typically passed on from *shifu* to apprentice in their daily lives. For a *shifu*, setting an example for the apprentices is a kind of Xiuxing in itself. New shamans who respect their *shifu* and their peers will often have more opportunities to perform rituals and will therefore be better able to improve their knowledge and skills about shamanism.

There have been some instances in which the *shifu* did not observe these principles. In one case, a *shifu* was so occupied playing mahjong that he or she neglected their apprentice's shaman illness. That *shifu* also tended to insult or condemn those who had differing opinions—not only apprentices, but also scholars and journalists. Many people ended up avoiding him.

With respect to finances, many Mongolian shamans quit their regular jobs to work full time; their salary comes from providing medical treatments, training apprentices, and performing rituals. There are no guidelines on how much money an apprentice or a client should pay the *shifu*[11], although the *shifu* usually asks for something reasonable that is within the means of an apprentice to pay. In one

---

10 Two *shifus*, Bao Haishan 包海山 and Wang Nuna 王努娜, had substantial differences of opinion. Therefore, they agreed to engage in a kind of competition where their apprentices would engage in trance and singing rituals. When Wang's apprentice successfully summoned an animal spirit, she was immediately woken from his trance, and very embarrassed to say to Bao that he had not learned yet. After the competition, they would have a friendly dinner. This kind of dialogue is positive. Yet it also suggests that, while the existence of animal spirits is accepted in Horqin shamanism, it is generally taboo to discuss this publicly, especially with other colleagues.

11 Typically, the *shifu* charges 100 or 200 yuan, though the shaman typically charges within the client's means to pay. Shaman rituals are time-consuming and labor-intensive. Afterwards, the *shifu*'s family many prepare a meal for as many as four to eight people. Many disciples live in the *shifu*'s house for an extended period of time, with the cost of living borne by the *shifu*. Some older shamans are unable to afford these costs, requiring the apprentice to pay approximately 500 yuan per month for living expenses. Some larger-scale activities, such as the *Oboo* 敖包 (*Oboo* is made up of sand and stone and was once used as a sign used to guide road and distinguish tribes) and ritual sacrifices, require a long time to prepare, as well as substantial funds to cover the venue, utensils, and other banquet materials. Thus, the *shifu* might ask for additional money from the apprentices. In some rituals, the apprentices may even be asked to provide the sacrificial lamb itself, or at least its equivalent in money.



case, a famous shaman named Don Huzler 唐呼日勒 had disciples from a wealthy family. After the *shifu* invited a spirit into the room, one of the apprentices took out a large roll of cash and placed it on the *shifu*'s desk. The *shifu* refused to accept it and even scolded the apprentice severely, claiming that such behavior would embarrass other disciples from less wealthy backgrounds. Such practice, the *shifu* insisted, was not conducive to healthy medical treatments or shamanistic practices.

Horchin shamanism has few rules for gender relations, but those rules that do exist tend to be related to issues of morality. Shamans of both sexes often live together in the *shifu*'s house. The *shifu* may also be asked to travel to the family of sick people in need of help. In such cases, the relationship between a man and a woman may cross a boundary of physical intimacy, leading to a rupture in the family or social stigmatization. A *shifu* named Wang Gen (pseudonym) had sexual relations with a female apprentice. The apprentice divorced her husband, and the family was broken up. The *shifu* lost his standing in the village as a result. Moreover, promiscuity is often associated with greed, leading some apprentices to take advantage of their power for greater wealth or sexual favors.

A *shifu* who is strict with themself has the moral authority to be strict with their disciples and admonish them when they break the rules. Don Huzler 唐呼日勒 had two apprentices, each with a family. Over time the two apprentices began to develop feelings for each other. However, Don Huzler severely admonished them and threatened to terminate his mentoring if they continued their behavior. The apprentices restrained themselves from then on. In addition, the *shifu* tried to stagger the apprentice's activities, having them perform in different locations so as to reduce their chances of meeting in the same place. For a shaman, maintaining high standards of virtue is essential for personal and family integrity, as well as for the reputation of the shaman's community more broadly. The success of shamans is, after all, dependent on public trust; when the community loses trust, the shaman's career is over.

Compassion is an essential part of Horchin shamanism. Many Horchin shamans are peasants from poor families, and some shaman apprentices are unable to hold regular jobs due to long-term illness. This may require the *shifu* to spend additional money on the apprentice's food and living expenses. The shaman must be able to sympathize with and care for the apprentices. The female shaman Qian Yulan 钱玉兰 is nearly 70 years old, has a large number of apprentices, and is proficient in Chinese and Mongolian. Many people, rich and poor, come to consult with her for guidance and healing or simply to visit her because of her fame. Regardless of their socioeconomic level, Qian is able to provide the same treatment to all visitors. In one instance, she was visited by a man who had caused an auto accident and was required to pay high medical expenses, worsening his already-difficult financial situation.

The man and his family came to the shaman's house for help. As they described the situation with tears in their eyes, the shaman showed empathy and tears of her own. She advised them on their situation. In addition, she gave them 500 yuan, thus increasing the gratitude felt by the family. A good shaman can assist the helpless and the mentally ill, alleviate their suffering, make them feel cared for, and give them the courage and confidence to overcome their difficulties. Not every shaman has this kind of compassion. And even if they do, it may not always last very long. Sustaining compassion over a long term requires daily practice, strengthened by a thorough understanding of the shaman's duties and a desire to help people with their problems.

## 5. The Shaman's Xiuxing in the Development of the Times

To this day, Mongolian people in the Horchin Region still rely heavily on shamanism. As a shaman cleric, it is important to use one's knowledge to serve the masses in their daily lives. This is the essence of modern Horchin shamanism, a responsibility that requires a strong calling and a lifelong commitment.

Over time, cultural and environmental changes have had an impact on the Xiuxing of the Horchin shaman. During our fieldwork we found that shamans have increasingly relied on brokers to locate

clients and apprentices[12]. One shaman surnamed Zhou 周 had an apprentice named Hai 海, whose mother died young and whose father was in prison for hurting other people. Hai had inherited the shaman's disease of his father, causing him considerable mental and physical pain, not to mention the death of his sheep. When he became Zhou's apprentice, he invited the spirits into his body while in a state of *sitügen* trance, and his mind and body soon returned to normal. He began showing greater diligence and self-discipline, and his quality of life improved as a result. Hai was a simple and honest man, accepting his *shifu*'s demands without question. However, Zhou only treated people who had been brought to him through a shaman broker, not daring to treat or deal with anyone else[13].

The contemporary Horchin shaman is also responsible for transmitting traditional shamanic culture, especially the spirit of traditional culture, such as a firm will to obey religious precepts and a charity that has persistently served the local people whether they are poor or rich. Another challenge faced with the changing times is that many old shamans who once clung to old traditions have begun to change their view toward certain taboos and requirements. For example, now that the government is making concrete steps to preserve intangible cultural heritage, many shamans have begun to break from the custom of only transmitting knowledge orally and have begun to write their knowledge down and seek to standardize that knowledge through government-issued certificates of authenticity[14]. It has taken the combined efforts of shamans and scholars to preserve the culture for the world to appreciate—but in such a way that its social role and significance for the Mongolian people do not get lost in tourist spectacles.

One example of this is the *boo* dance, which is now performed for large audiences of tourists and journalists. It may also be performed at rituals and religious ceremonies, such as the Okubo 敖包 and Guoguan 过关. This dance, which was once performed only when a client had a problem and a spirit needed to be summoned, is now performed for large crowds for entertainment purposes, not to summon a spirit. This trend, of performing shaman rituals for show and not for healing, conflicts with the requirements for Xiuxing, since shamans are no longer able to treat the rituals as sacred traditions. Xiuxing, which is typically achieved through religious practice, can be lost if the rituals are performed as a tourist spectacle. Through these efforts, the state and the public have gradually reaccepted the view that Horchin shamanism is not feudal superstition but is instead an important part of Mongolian cultural heritage. The dances and religious doctrines of Horchin shamanism have been included on the official intangible cultural heritage list, while at the same time the shamans have been able to continue engaging in their activities for purposes of serving clients and healing real ailments.

There are still many questions that have to be answered regarding the adaptation of Horchin shamanic culture to the realities of the twenty-first century. This requires innovation and creativity. Horchin shamans have an obligation and responsibility to pass on their culture. This is not an easy task, as it requires them to make a lifelong commitment to helping the wider world develop a deeper understanding of Horchin shamanic culture.

---

[12] There are local "brokers" in the Horchin Region who specialize in connecting interviewers and shamans, earning money from both sides.

[13] The apprentice has both the "black boo" 黑博 (the shaman not affected by Lamaism) *sitügen* and the "Lai Qing" 莱青 (the shaman affected by Lamaism deeply) *sitügen* tourist spectacle. The disciple's body movements during the ritual were broad with quick rotations. He would beat on three drums with one hand or six drums with two. His dances were exemplary, and he would be the mentor for a younger generation of shamans. His *shifu* would often ask him to perform for guests. If he was in his own home, he would have to prepare dishes according to the *shifu*'s requirements. Initially, the villagers were reluctant to allow us as researchers to visit. Apparently, this was because the young shaman spent much time and energy practicing his *shifu*'s instructions but was not improving his own quality of life. They did not understand why someone would want to interview someone with these defects.

[14] In light of government preservation efforts in recent years, the Mongolian shaman *boo* dance has been given increased attention. According to a recent survey, shamans are busily applying for cultural certificates of intangible cultural heritage to take advantage of the economic benefits and greater reputation that these certificates could bring. Shamans are thus strongly encouraged to cultivate relationships with scholars, students, journalists, government officials, and even travel agencies. Moreover, possessing such certificates of cultural expert gives the shaman greater prestige, resulting in more invitations to public performances, more contacts in general, and more shamanic hopefuls wanting to be their apprentices. Shamans without such certificates have more difficulty achieving these things.

## 6. Conclusions

Based on the above field data, we see that insights into the internal mechanisms for motivation and acquisition of religious knowledge are critical to the study of how believers find their way to transcendence. However, this data also illustrates the various externally observable daily practices that are important for the process of physical and spiritual transformation from ordinary person to religious specialist.

Certain daily habits are necessary for a shaman to achieve their Xiuxing. Such habits include respect for the *shifu*, commitment to fulfilling a shaman's obligations and responsibilities, obedience to religious principles, cultivation of higher morality, compassionate care for others, and acceptance of the mission to transmit and inherit shamanic culture. In addition to serving the beings in the spirit world through rituals of animal sacrifice, it is important to serve the people in the real world through rituals of healing. Even the *sitügen* trance should be used to serve real people.

**Author Contributions:** This paper is completed on the basis of our common fieldwork since 2013. we defined the framework of the paper and the scope of the quoting material data together. Yumin Lun finished the writing of the paper, Xiaomei Dong proofread it and revised it.

**Funding:** This research was funded by [the Humanities and Social Sciences Foundation of the Ministry of Education of China] grant number [17YJC730006] (中国教育部人文社科青年基金项目"科尔沁蒙古萨满文化传承人口述资料发掘、整理与研究": 17YJC730006).

**Conflicts of Interest:** We declare that there is no conflict of interests regarding the publication of this article.

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
