# Peer review of "A Study on the Xiuxing of Contemporary Horchin Mongolian Shamanism"

_religions, doi:10.3390/rel10020112_

Round 1
Reviewer 1 Report
Comments on the Mongolian Shamanism article
A more informative statement of methodology should appear early in the article. It should discuss the number of persons that served as the main sources of information, Line 83 refers to “more than 100” people interviewed. What were the questions posed? Were the same questions posed to all people? What was the language of the interviews? Did the respondents know Mandarin? Did the researchers know Mongolian? How were the interviews recorded: electronically or in written notes?
Lines 22 – 25. The importance of xiuxing is an important finding that deserves analysis. It is being discussed appropriately here. But it should NOT be used repeatedly in an English-language article. An adequate English translation miust be found. Underlying all three reported definitions of xiuxing is the literal meaning of “repairing one’s conduct” or ”rectifying one’s behavior”. (xiu = repair or fix; xing here = behavior.) Perhaps English terms like “religious discipline” or “spiritual progress” could be used as a translation.
Examples:
Line 70. “However, there are well-established requirements for a shaman to achieve Xiuxing.”
Alternative “However, there are well-established requirements for a shaman to demonstrate spiritual progress”.
Lines 75-77. It would be difficult for ordinary people to achieve Xiuxing … on their own.
Alternative: It would be difficult for ordinary people to learn ritual skills and to advance morally… on their own.
Line 194. “The most important condition for Xiuxing is for the apprentice to learn with the Dshifu.”
Alternative: The most important condition for advancing spiritually is for the apprentice to learn with the shifu./ (Note: shifu (“master father”) can be used, as it has a clear definition. Xiuxing is abstract and should be translated into clear English, depending on the specific context of the sentence. .
Also: if the term xiuxing is occasionally used, it should be in lower case and italicized.
Also: the anglophone reader may assume that the “x” is pronounced as the “x” in “six”. You might add a footnote: the word is pronounced like syo-sying or sho-shing.
Line 81. Sitügen. The belief is interesting. It appears to be a type of “tutelary spirit” that adopts and guides an individual. Some questions that an anthropologist would like to know: Is it an anthropomorphic spirit in human form or a theriomorphic (animal-like) spirit such as a longwang? Do all humans have a sitügen or only shamans? Are there both male and female variants. Can a male have a female situgen, and vice versa? Because the word is strange to anglophone ears, the authors might consider translating it as “spirit guide” or “guardian spirit” rather than continuing to use the term situgen.
Line 106. The question of “attachment / possession”. The interesting phrase. 附体状态开口说话 might be explained to the anglophone reader. It literally means a “possession state (where a spirit) opens the mouth and speaks)..” The individual goes into a trance as a spirit possesses his body and opes his mouth to speak through it. The common English term is “possession”, which the authors correctly use, rather than “attachment”, the literal meaning of 附. (fu).
Line 198. Healing is mentioned. Good. It will be important at some point to state explicitly what the major service which the Horqin shamans provide to their clients, the ones that do not want to become shamans. Is healing the major role? Are there other services which they provide? Weather management? Finding lost articles? Determining favorable dates for weddings or house building?
Line 204. Explain shu or put it in English.
Line 206: “spiritual progress” rather than Xiuxing.
Line 211. Temples are mentioned. Is a shaman’s services associated with a particular temple? Or does he work mostly from his home or from the home of his clients?
Line 244. Could be rephrased, or explained. “Shamans not only have to know the rituals, but they are expected to advance in moral virtue as well.”
Line 275. Mention of a “female apprentice”. This means that women can be shamans and perhaps even shifu as well. That should be discussed at least briefly. And that presents a linguistic dilemma in English (less so in Mandarin). The text has been talking constantly about “he”, “his”, “him” when referring to shifu and apprentices, implicitly excluding women. It is a challenge in English to find gender-neutral terms when a group has both men and women. One way of handling it is to use the plural. .Instead of saying “An apprentice has to respect his shifu”, one could say “apprentices have to respect their shifu”.
Lines 325-333. Very important section: the potentially deleterious impact of performances for tourists on genuine shamanic practices. When the government supports Cultural Heritage, it is referring to songs, dances, clothing, embroidery, legends, folk tales, and the like. The government of the PRC does not support or encourage shamanic healing, which is the major role of the shaman. This is a major dilemma that deserves discussion with respect to the revival of Chinese shamanism.
Line 334 – 338. The paragraph raises two controversial issues which merit discussion.
1, The common rejection of shamanism as “superstitious” because of its reliance on spirits and rituals to heal illness.
2. The danger that performances as tourist spectacles will somehow be harmful to the shamanic system.
With respect to (1), rejection of shamanic healing is now strong in China. When given the choice, most Chinese (including anthropologists), whether Han or minority minzu, will go to physicians or TCM healers, not shamans, when they or their children get ill. Is there something wrong with that choice? Is the argument being made that Horqin Mongolians should continue to frequent shamans for healing services? Presumably not. So therefore, please specify what aspect of “traditional shamanic culture” should be preserved? This is a sensitive, complicated question that deserves a full length article. It can be mentioned briefly here.
Proposition 2 – the danger that tourist spectacles will produce phony shamans and phony possession – is also accurate. That also deserves lengthy analysis.
Comments on the title and the abstract.
Improved title: “The recruitment and training of shamans: A case study from Inner Mongolia”
The word “xuexing” is foreign to 99% of anglophone readers. On the anglophone web it is usually used as a person’s name. It confuses the title. The proposed alternative title uses plain English to capture the major features of the article.
The abstract (lines 5 – 14) needs radical improvement. It begins with a discussion of origins and historical linkages which are tangential or irrelevant to the major findings. The abstract should have the following elements:
1. A statement of the research purpose. “Research was carried out on the procedures for recruiting and training shamans.among the …..etc.”
2. A statement of the time frame and data gathering methodology of the research.
3. A statement of the major findings in the form of 1. Xxxx 2.xxxx 3xxxxx.
4. A brief statement of the potential relevance of the research.
Lines 11-13. “The primary challenge currently facing Horqin shamans is the development of a new religious concept that conforms to traditional religious practices.” That is vague and unintelligible. Perhaps reword “We also found that Horqin shamans are struggling to adapt their traditional religious practices to the belief systems of the modern Chinese world.”
Lines 13-14. “Only a shaman with compassion and high moral character can maintain this career over the long term.” The article does not contain data to justify that statement. A statement of noble ideals is being incorrectly presented as a statement of empirical fact.
The researchers did encounter a belief about the desired qualities of a shaman. The abstract could therefore legitimately say “We encountered a belief in the region that a successful shamanic career presupposes not only knowledge of rituals, but also compassionate and principled behavior with respect to the clients and the community.” That is an ideal belief that can be reported. It is, however, not a proven empirical fact that most shamans behave according to the ideal. It is best reported as an ideal, not as an empirical fact concerning the behavior of shamans.
To conclude: The article has many new insights that merit publication.
Author Response
Response to Reviewer 1 Comments
Thank you very much for you very insightful and extremely helpful comments.
Point. 1
A more informative statement of methodology should appear early in the article. It should discuss the number of persons that served as the main sources of information, Line 83 refers to “more than 100” people interviewed. What were the questions posed? Were the same questions posed to all people? What was the language of the interviews? Did the respondents know Mandarin? Did the researchers know Mongolian? How were the interviews recorded: electronically or in written notes?
Response. 1
The original title we wanted to do was "The Growth process of the New Horchin Shaman." Since 2013, we have been studying the questions like the growth and maturity of shamans and the transformation of their identities. Through our fieldwork, we tracked and interviewed more than 100 shamans and prospective shamans and obtained lots of fieldwork data, most of whom spoke Mandarin, and some who only spoke Mongolian, and we made audio and video recordings. With the help of Ph.D Zhao Furong, a Mongolian scholar, these materials were translated into Chinese. We also interviewed these people’s family, friends and neighbors, as well as well-known researchers.
Ponit.2
Lines 22 – 25. The importance of xiuxing is an important finding that deserves analysis. It is being discussed appropriately here. But it should NOT be used repeatedly in an English-language article. An adequate English translation miust be found. Underlying all three reported definitions of xiuxing is the literal meaning of “repairing one’s conduct” or ”rectifying one’s behavior”. (xiu = repair or fix; xing here = behavior.) Perhaps English terms like “religious discipline” or “spiritual progress” could be used as a translation.
Examples:
Line 70. “However, there are well-established requirements for a shaman to achieve Xiuxing.”
Alternative “However, there are well-established requirements for a shaman to demonstrate spiritual progress”.
Lines 75-77. It would be difficult for ordinary people to achieve Xiuxing … on their own.
Alternative: It would be difficult for ordinary people to learn ritual skills and to advance morally… on their own.
Line 194. “The most important condition for Xiuxing is for the apprentice to learn with the Dshifu.”
Alternative: The most important condition for advancing spiritually is for the apprentice to learn with the shifu./ (Note: shifu (“master father”) can be used, as it has a clear definition. Xiuxing is abstract and should be translated into clear English, depending on the specific context of the sentence. .
Also: if the term xiuxing is occasionally used, it should be in lower case and italicized.
Also: the anglophone reader may assume that the “x” is pronounced as the “x” in “six”. You might add a footnote: the word is pronounced like syo-sying or sho-shing.
Response. 2
Scholars who initially advocated the Xiuxing anthropology also wanted to use words such as religious practice instead of xiuxing. The use of Pinyin to express it is the opinion of Professor Robert P. Weller of Boston University. In fact, the meaning of this word includes the both mean of ‘religious discipline’ and ‘spiritual progress’ that from your opinion. Using it to study religious believers is a relatively new research path, and this article is also an attempt. I have adopted most of your opinions.
Point. 3
Line 81. Sitügen. The belief is interesting. It appears to be a type of “tutelary spirit” that adopts and guides an individual. Some questions that an anthropologist would like to know: Is it an anthropomorphic spirit in human form or a theriomorphic (animal-like) spirit such as a longwang? Do all humans have a sitügen or only shamans? Are there both male and female variants. Can a male have a female situgen, and vice versa? Because the word is strange to anglophone ears, the authors might consider translating it as “spirit guide” or “guardian spirit” rather than continuing to use the term situgen.
Response. 3
Sitügen is a local term of the Horchin shamanism. Generally, sitügen has several types: a),the souls of the shamans’ own ancestors who was before as a shaman, after he or she died, their soul would find a person from his or her descendants. Most of them would find their descendants with the same gender. This type of sitügen is their main deity, and the shamans often through a special ceremony to request their sitügen leave away from the place of residence by the possession to help them handle some things. b) Some sitügen would find person without blood relationship. c) Sometimes the animal spirts of the shamans such as fox, white snake, mouse, etc, also be called as sitügen. Most of the time the local shamans use this word to refer to the personality spirt of their own.
Point 4
Line 106. The question of “attachment / possession”. The interesting phrase. 附体状态开口说话 might be explained to the anglophone reader. It literally means a “possession state (where a spirit) opens the mouth and speaks)..” The individual goes into a trance as a spirit possesses his body and opes his mouth to speak through it. The common English term is “possession”, which the authors correctly use, rather than “attachment”, the literal meaning of 附. (fu).
Response 4 . I fully agree with you.
Point 5.
Line 198. Healing is mentioned. Good. It will be important at some point to state explicitly what the major service which the Horqin shamans provide to their clients, the ones that do not want to become shamans. Is healing the major role? Are there other services which they provide? Weather management? Finding lost articles? Determining favorable dates for weddings or house building?
Response 5.
Horqin shamans offer the same services to potential shamans who do not want to be shamans, as those who want to be shamans. The difference is shifu teach different knowledge and asked them what to do between the different attitude people. For the ones that do not want to become shamans, their shifu usually ask them must to sacrifices their sitügen, such as in the time lunar calendar 7, July and 9 Sep, never forget to do that.
The services provided by Horqin shaman to the public mainly include sacrificial rituals, treatments, removing evil, praying for blessings, etc. Now Sacrificial rituals for rain, is their mainly active about weather management. They do not do some things like finding lost articles of determining favorable dates for weddings or house building. These skills need to be learned from the Han, which Horqin shamanism does not.
Point 6.
Line 204. Explain shu or put it in English.
Line 206: “spiritual progress” rather than Xiuxing.
Line 211. Temples are mentioned. Is a shaman’s services associated with a particular temple? Or does he work mostly from his home or from the home of his clients?
Line 244. Could be rephrased, or explained. “Shamans not only have to know the rituals, but they are expected to advance in moral virtue as well.”
Response 6 .
Line 204 The meaning of shu is skill, method.
Line 206: I fully agree with you.
Line 211. It is a transformation mistake, I have changed.
Line 244. I used The religion moral (or Tao) Xiuxing as A Shaman instead.
Point 7.
Line 275. Mention of a “female apprentice”. This means that women can be shamans and perhaps even shifu as well. That should be discussed at least briefly. And that presents a linguistic dilemma in English (less so in Mandarin). The text has been talking constantly about “he”, “his”, “him” when referring to shifu and apprentices, implicitly excluding women. It is a challenge in English to find gender-neutral terms when a group has both men and women. One way of handling it is to use the plural. .Instead of saying “An apprentice has to respect his shifu”, one could say “apprentices have to respect their shifu”.
Response7.
Yes, women can be shamans and perhaps even shifu as well. This state has a long history, and so it is today. I used they and their instead.
Point 8.
Lines 325-333. Very important section: the potentially deleterious impact of performances for tourists on genuine shamanic practices. When the government supports Cultural Heritage, it is referring to songs, dances, clothing, embroidery, legends, folk tales, and the like. The government of the PRC does not support or encourage shamanic healing, which is the major role of the shaman. This is a major dilemma that deserves discussion with respect to the revival of Chinese shamanism.
Response 8.
Yes, I fully agree with you. I am collecting information and preparing to write on this subject.
Point 9.
Line 334 – 338. The paragraph raises two controversial issues which merit discussion.
1, The common rejection of shamanism as “superstitious” because of its reliance on spirits and rituals to heal illness.
2. The danger that performances as tourist spectacles will somehow be harmful to the shamanic system.
With respect to (1), rejection of shamanic healing is now strong in China. When given the choice, most Chinese (including anthropologists), whether Han or minority minzu, will go to physicians or TCM healers, not shamans, when they or their children get ill. Is there something wrong with that choice? Is the argument being made that Horqin Mongolians should continue to frequent shamans for healing services? Presumably not. So therefore, please specify what aspect of “traditional shamanic culture” should be preserved? This is a sensitive, complicated question that deserves a full length article. It can be mentioned briefly here.
Proposition 2 – the danger that tourist spectacles will produce phony shamans and phony possession – is also accurate. That also deserves lengthy analysis.
Response 9.
I rewrote this part. This question full with academic value. We think that the traditional shamanistic culture is relatively clear to the shamans, and this article also explains that. For ordinary people, it is mainly its outstanding ideas, such as the concept of ecological protection, the equality of all living beings, love others and respect teachers, etc.
Point 10.
Comments on the title and the abstract.
Improved title: “The recruitment and training of shamans: A case study from Inner Mongolia”
The word “xuexing” is foreign to 99% of anglophone readers. On the anglophone web it is usually used as a person’s name. It confuses the title. The proposed alternative title uses plain English to capture the major features of the article.
The abstract (lines 5 – 14) needs radical improvement. It begins with a discussion of origins and historical linkages which are tangential or irrelevant to the major findings. The abstract should have the following elements:
1. A statement of the research purpose. “Research was carried out on the procedures for recruiting and training shamans.among the …..etc.”
2. A statement of the time frame and data gathering methodology of the research.
3. A statement of the major findings in the form of 1. Xxxx 2.xxxx 3xxxxx.
4. A brief statement of the potential relevance of the research.
Lines 11-13. “The primary challenge currently facing Horqin shamans is the development of a new religious concept that conforms to traditional religious practices.” That is vague and unintelligible. Perhaps reword “We also found that Horqin shamans are struggling to adapt their traditional religious practices to the belief systems of the modern Chinese world.”
Lines 13-14. “Only a shaman with compassion and high moral character can maintain this career over the long term.” The article does not contain data to justify that statement. A statement of noble ideals is being incorrectly presented as a statement of empirical fact.
The researchers did encounter a belief about the desired qualities of a shaman. The abstract could therefore legitimately say “We encountered a belief in the region that a successful shamanic career presupposes not only knowledge of rituals, but also compassionate and principled behavior with respect to the clients and the community.” That is an ideal belief that can be reported. It is, however, not a proven empirical fact that most shamans behave according to the ideal. It is best reported as an ideal, not as an empirical fact concerning the behavior of shamans.
Response 10. I've modified them done.
Reviewer 2 Report
‘The Study on the ‘Xiuxing’ of Modern Horqin Mongolian Shamanism’ details some aspects of the contemporary practice of Horqin shamanism, arguing that Xiuxing (daily, self-cultivation of certain practices and moral subjectivities) is central to developing on a shamanic path. While the argument is ambitious for the relatively limited word count, it achieves the basic aim of describing ritual practices and the contemporary economic and health-related circumstances under which becoming a shaman is made compelling to people, along with the associated challenges. It also successfully paints a convincing ethnographic portrait of general discursive trends in the ways shamanism is understood locally and the, at times, tigh rope-like ‘rules’ by which correct shamanic practice ‘ought’ take place. As the reader progresses through the argument, it becomes more and more clear that the author has spent quite a good deal of time with shamanic interlocutors, which is important to making claims based on participant-observation. One of the great strengths of the article is the connecting to the political economy, even if sparingly and tangentially done. The tension created between a formerly-banned practice to one encouraged under the framework of Intangible Cultural Heritage is a theme I would recommend the author pull out, and apply ethnographic example and analysis.
However, there are substantial improvements that need to be made structurally, both in terms of the argument itself and layout. There are also citations that should be added, as well as concepts and themes employed as if they were ‘common sense’ that need explanation and justification for the piece to work as a whole. I will address these in turn, loosely progressing from beginning to end.
Line 2: While the title is succinct and informative, I would ask that the author consider using ‘A Study…’ as opposed to ‘The Study…’. Crucially, what is meant by modern? Would ‘contemporary’ be more appropriate?
The abstract needs work. While on paper the elements inside the article appear in the abstract, clarification is needed. The way it reads now, it is as if Xiuxing is a subset of shamanic practice, which, as I understand having read the opening paragraphs, is clearly not the case. Is the ‘new religious concept’ (line 12) Xiuxing? Unclear. An assumption about the ‘uncut’ line of ‘traditional’ shamanism is raised in the 13th line of the abstract which remained for me throughout an underdeveloped and assumed reality of Horqin shamanism. Is the author interested in making a distinction between tradition and modern (as suggested by the title)? If so, substantial work in the literature review is needed.
Line 22: I would consider removing the ‘cultural’ in ‘cultural anthropology’ as, the way the sentence reads now, there is an anachronistic feel. The structural-functionalism this sentence references (references very much needed) in many ways precedes ‘cultural anthropology’ as a discipline.
Lines 25-26: citation needed. Who has coined and used ‘Xiuxing anthropology’?
Paragraphs beginning with line 27 and 37 need restructuring. There is a lack of coherence, and it is very unclear as to what exactly is being argued. The author seems to make the same point again and again over the course of several paragraphs that there are institutionalized and individual aspects to Xiuxing, a concept which pertains to ethical self-formation typically associated with religiosity.
Line 52: There is a sudden jump to shamans that needs to be flagged up. It might be more appropriate to remove mention in that paragraph and wait for the following to bring it up more substantially.
Lines 53-54: Is the author promoting Jinguo’s split between religious worlds and daily life? Surely the point of the Chinese scholars’ call to develop an anthropology of Xiuxing is to illustrate the ways in which these realms are not held apart but deeply integrated.
Line 64: ‘We’ and ‘my fieldwork’ (line 90) are incompatible.
Line 71: ‘Traditional Horqin’. What is this? I found myself throughout the article wondering what aspects of ‘tradition’ (which needs addressing in its own right) belong to Horqin shamanism and what aspects belong to Horqin sociality more generally? This needs clarifying, and would actually help to tease out the subtle theme of shamans as ‘cultural heritage protectors’ that I think is really interesting (raised again Line 227 with shamans ‘deliberately studying their culture’ and Line 323 with ‘preserving Intangible Cultural Heritage’). This would be an interesting theme to explore in more depth, as it links the practice up with state-level and international discourses on cultural heritage, which seem to influence the form the practice itself takes.
Line 82 and footnote 2: Shamanic illness here is portrayed here as if it is a requirement. If so, if as stated in footnote 2, ‘they must experience mental and physical problems’, we need evidence of this. The author simply telling us (but making an extremely bold claim in the process) is not enough. To this end, the author should review Uranchimeg Ujeed’s work generally and in particular the 2018 “Becoming Shamans to be Healed: Reasons for Contemporary Proliferation of Horchin Shamanism” in Acta Orientalia 4. There is also a missing citation in footnote two.
Once the argument gets to page 3, the Xiuxing aspect is somewhat drowned in detailing of shifu relations and shamanic illness. Although fascinating, the reader is not sure why it’s relevant to the argument itself. What is the relation of shifu-disciple relationship to inner worlds?
Line 149-150: Examples of financial strain of shamanic rituals can also be found in Manduhai Buyandelger’s work, which might be worth citing.
Also on page 4, the reader isn’t yet seeing the ‘inner worlds’ of shamans. Although becoming a shaman for treatment of illness and/or cases of economic misfortune is/are detailed, we don’t have any intra-personal insight that one would expect from ethnographic methodology.
178: What are the ‘ordinary circumstances’ for becoming a shaman? Surely the widespread notion of shamanic illness, cited by the author earlier, would refute this point?
Lines 181 and 79: It’s not clear why the subsections ‘The Xiuxing way as A Shaman’ and ‘The Xiuxing Way to Become A Shaman’ are labeled as such. Does the author intend to walk the reader through such a progression? If so, it might be clearer to ethnographically ground the progression with one or two interlocutors with whom the reader could relate.
Line 189: ‘traditional Inner Mongolian society’: what is this? If the author is drawing on particular work, it needs to be cited.
Line 209-10: the scholars, government officials and shamans that are helping to preserve Horqin shamanism is a really interesting strand, and the tension produced between that and former state-sanctioned bans of shamanism as ‘backwards’ represents a tension that should be pulled out. I would recommend pulling footnote 7 into the text and engaging this fascinating dimension more substantially. The first line of footnote 7 fails to take into account vibrant Darkhad and Buriat shamanic traditions, which scholars have argued remained to a certain extent distinct from Buddhism.
Line 244: Consider revising this heading. A suggestion could be: ‘Strengthening the Xiuxing of Tao’
Line 288: Is the author considering shamanism a religion? This seems to be implied but needs substantial engagement.
By Line 304, the author has mentioned the importance of daily practice but we have seen no illustration of this. What are the daily practices of shamans?
Lines 329-333: the conflict between shamanism as a tourist spectacle and adhering to the requirements of Xiuxing: A fascinating tension that could really be developed. It would be really great if the author could illustrate how shamans manage such conflicts and inhabit the everyday space between reified spectacle and cultivated ‘inner world’.
Paragraphs beginning with line 334 and 339 need to be integrated. If not, the former sounds extremely outdated and jars with the rest of the article.
Line 346: Is it that Horqin shamans have an obligation or feel that they have an obligation?
Line 350: What are the ‘internal mechanisms’? We see very little of the internal lives of the interlocutors.
Line 355: What are the daily habits? Again, example/evidence is not provided.
361-362: The final sentence is completely unsuitable.
Author Response
Response to Reviewer 2 Comments
Thank you very much for your very insightful and extremely helpful comments.
Point.1
Line 2: While the title is succinct and informative, I would ask that the author consider using ‘A Study…’ as opposed to ‘The Study…’. Crucially, what is meant by modern? Would ‘contemporary’ be more appropriate?
Response.1
Yes, I have used contemporary to instead of modern.
Point.2
The abstract 2needs work. While on paper the elements inside the article appear in the abstract, clarification is needed. The way it reads now, it is as if Xiuxing is a subset of shamanic practice, which, as I understand having read the opening paragraphs, is clearly not the case. Is the ‘new religious concept’ (line 12) Xiuxing? Unclear. An assumption about the ‘uncut’ line of ‘traditional’ shamanism is raised in the 13th line of the abstract which remained for me throughout an underdeveloped and assumed reality of Horchin shamanism. Is the author interested in making a distinction between tradition and modern (as suggested by the title)? If so, substantial work in the literature review is needed.
Response.2
I rewrote the abstract.
Point.3
Line 22: I would consider removing the ‘cultural’ in ‘cultural anthropology’ as, the way the sentence reads now, there is an anachronistic feel. The structural-functionalism this sentence references (references very much needed) in many ways precedes ‘cultural anthropology’ as a discipline.
Lines 25-26: citation needed. Who has coined and used ‘Xiuxing anthropology’?
Paragraphs beginning with line 27 and 37 need restructuring. There is a lack of coherence, and it is very unclear as to what exactly is being argued. The author seems to make the same point again and again over the course of several paragraphs that there are institutionalized and individual aspects to Xiuxing, a concept which pertains to ethical self-formation typically associated with religiosity.
Response.3
The first two modifications have been completed.
Paragraphs beginning with line 27 and 37, in these paragraphs, I have combed the main views of domestic scholars on the anthropology of practice, which is a new attempt to expand religious education and anthropology research path through Chinese local cultural resources. The connotation, extension and application of the theory of Xiuxing are still developing. This part, I have not made major changes.
Point.4
Line 52: There is a sudden jump to shamans that needs to be flagged up. It might be more appropriate to remove mention in that paragraph and wait for the following to bring it up more substantially.
Lines 53-54: Is the author promoting Jinguo’s split between religious worlds and daily life? Surely the point of the Chinese scholars’ call to develop an anthropology of Xiuxing is to illustrate the ways in which these realms are not held apart but deeply integrated.
Response.4
I have changed these points.
Lines 53-54: Yes, it is one purpose of Chinese scholars call to develop anthropology of Xiuxing. This part exists some logical problem, so I delete some sentients.
Point.5
Line 64: ‘We’ and ‘my fieldwork’ (line 90) are incompatible.
Response.5
I've modified this and related issues.
Point.6
Line 71: ‘Traditional Horchin’. What is this? I found myself throughout the article wondering what aspects of ‘tradition’ (which needs addressing in its own right) belong to Horchin shamanism and what aspects belong to Horchin sociality more generally? This needs clarifying, and would actually help to tease out the subtle theme of shamans as ‘cultural heritage protectors’ that I think is really interesting (raised again Line 227 with shamans ‘deliberately studying their culture’ and Line 323 with ‘preserving Intangible Cultural Heritage’). This would be an interesting theme to explore in more depth, as it links the practice up with state-level and international discourses on cultural heritage, which seem to influence the form the practice itself takes.
Line 178: What are the ‘ordinary circumstances’ for becoming a shaman? Surely the widespread notion of shamanic illness, cited by the author earlier, would refute this point?
Line 189: ‘traditional Inner Mongolian society’: what is this? If the author is drawing on particular work, it needs to be cited.
Response.6
Three external audit experts, including you, mentioned these two questions. The study of the history of traditional Horchin or traditional Horchin shamanism is also an important issue, which involves the question of traditional definition and the question of distinguishing traditional and contemporary standards. A special article is needed to deal with this problem and I have made appropriate modifications to this paper. In the past, I have always regarded the shaman recorded in the article "A Preliminary Study of the Art of Horchin Boo" in Tongliao Bai Cuiying, China, as the last group of shamans that grew up in the traditional society. They were mainly active before the Cultural Revolution. After the Cultural Revolution, old shamans such as Xue Renqin were still engaged in shamanism, but their disciples, including the oldest, were active even later after China reform and opening up, and I usually regarded them as the first generation of shamanies in contemporary society. Many of their thoughts and ways of behavior carry the brand of contemporary society. But there was no detailed analysis or discussion of how I distinguished myself. I will carry out further research in the future. The question of the relationship between non-posthumous protection and the Horchin shamanism may be an inevitable and important issue in the study of shamanism today, and in my opinion, there is tension, a game, and a fusion between the two. I have begun to write on related issues.
Point 7.
Line 82 and footnote 2: Shamanic illness here is portrayed here as if it is a requirement. If so, if as stated in footnote 2, ‘they must experience mental and physical problems’, we need evidence of this. The author simply telling us (but making an extremely bold claim in the process) is not enough. To this end, the author should review Uranchimeg Ujeed’s work generally and in particular the 2018 “Becoming Shamans to be Healed: Reasons for Contemporary Proliferation of Horchin Shamanism” in Acta Orientalia 4. There is also a missing citation in footnote two.
Line 149-150: Examples of financial strain of shamanic rituals can also be foundwhich might be worth citing.
Response 7.
In this article, we are not only wrote about shamanic illness, but also traffic accidents but personal safety, loss of property accident, and so on. In our fieldwork, most potential shamans become shamans, mostly because of shamanic illness.
Footnote 2 ,they must, I have changed to most of them, and added some arguments. When we work in the field, we always ask every shaman the reasons they contact with shamanism and become shaman, and most people start with these accidents, especially if they get some diseases that cannot be cured by medicine. Most people go to the hospital after they get sick, but they think contemporary medicine is ineffective for their symptoms, and after some twists or turns, they turn to shamans. There are also some people who have symptoms and go straight to shaman. I tried to ask them for some hospital cases, diagnostics and so on, and they could not provide them to us. Unfortunately, we have not gone to some of Horchin hospitals to interview doctors about the phenomenon.
Uranchimeg Ujeed is a very respectable scholar, and many Mongolian scholars often mention her, but I can only find a summary of this article, I commissioned friends abroad to help, but they have not replied to me. I will continue to look for this document. And the same thing with Manduhai Buyandelger’s work,.
Point 8.
Once the argument gets to page 3, the Xiuxing aspect is somewhat drowned in detailing of shifu relations and shamanic illness. Although fascinating, the reader is not sure why it’s relevant to the argument itself. What is the relation of shifu-disciple relationship to inner worlds?
Also on page 4, the reader isn’t yet seeing the ‘inner worlds’ of shamans. Although becoming a shaman for treatment of illness and/or cases of economic misfortune is/are detailed, we don’t have any intra-personal insight that one would expect from ethnographic methodology.
Response 8.
I fully agree with you. I have added a simple example to footnote 6.
Point.9
Lines 181 and 79: It’s not clear why the subsections ‘The Xiuxing way as A Shaman’ and ‘The Xiuxing Way to Become A Shaman’ are labeled as such. Does the author intend to walk the reader through such a progression? If so, it might be clearer to ethnographically ground the progression with one or two interlocutors with whom the reader could relate.
Response .9
Yes. We also follow this process, but the related fieldwork is continue, and the xiuxing of some apprentices mention at this article is part of this process. Our next study, "the growth of the new Shaman," will focus on this issue.
Point.10
Line 209-10: the scholars, government officials and shamans that are helping to preserve Horchin shamanism is a really interesting strand, and the tension produced between that and former state-sanctioned bans of shamanism as ‘backwards’ represents a tension that should be pulled out. I would recommend pulling footnote 7 into the text and engaging this fascinating dimension more substantially. The first line of footnote 7 fails to take into account vibrant Darkhad and Buriat shamanic traditions, which scholars have argued remained to a certain extent distinct from Buddhism.
Response 10.
I have put the footnote in the proper position of the text and added Inner Mongolia as the qualifiers at the beginning .
Point.11
Line 244: Consider revising this heading. A suggestion could be: ‘Strengthening the Xiuxing of Tao’
Response .11
I have used The religion moral (or Tao) Xiuxing as A Shaman instead.
Point.12
Line 288: Is the author considering shamanism a religion? This seems to be implied but needs substantial engagement.
Response.12
This is mainly about the concept of religion, and Chinese scholars have been arguing about it. In view of the importance of this academic term and its low relevance to this paper, I have deleted this sentence.
Point.13
By Line 304, the author has mentioned the importance of daily practice but we have seen no illustration of this. What are the daily practices of shamans?
Line 355: What are the daily habits? Again, example/evidence is not provided.
Response 13.
In addition to the specific day of sacrifice, most of the religious activities of Horchin shaman are in the ordinary daily life, and also with the depth of the daily life of the Horchin Mongols. In this article, we usually refer to religious practice in Shaman’s learning process, while serving the public is in daily practice.
Point.14
Lines 329-333: the conflict between shamanism as a tourist spectacle and adhering to the requirements of Xiuxing: A fascinating tension that could really be developed. It would be really great if the author could illustrate how shamans manage such conflicts and inhabit the everyday space between reified spectacle and cultivated ‘inner world’.
Response.14
Three reviewers, including you, all have talked about this issue. It is a important issue for the shamans. I have set out to write a special article on this subject.
Point.15
Paragraphs beginning with line 334 and 339 need to be integrated. If not, the former sounds extremely outdated and jars with the rest of the article.
Response.15
I have rewrote this paragraphs,
Point.16.
Line 346: Is it that Horchin shamans have an obligation or feel that they have an obligation?
Response.16
Some of them have this consciousness .
Point.17
Line 350: What are the ‘internal mechanisms’? We see very little of the internal lives of the interlocutors.
Response.17
This may not be specific enough, I think the interaction between shifu and apprentice is an internal mechanism.
Point.18
361-362: The final sentence is completely unsuitable.
Response.18
I have deleted them.
Reviewer 3 Report
See attached file. Contact me if the attached file has any issues.
Thanks!

Author Response
Response to Reviewer 3 Comments
Thank you very much for your very insightful and extremely helpful comments.I have corrected the spelling problem you have pointed out, so these problem I will not respond to you.
Point.1
The author could do more to represent the historic and current context of these practices. As I was reading the article, I was wondering about the relationship between apparent encouragement of these practices and the current “morality” drive of the CCP. To what degree are they inter-related? The attacks on religion post-1949 have to be addressed in the body of the paper, not in a footnote.
Response.2
I think this is a very complex problem, a short article cannot to explain it clearly. But through our fieldwork, we can be sure that shamanism itself has a strong moral tendency. What’s more China government institutions, especially cultural institutions, do not mean that all official are the CCP, also include democratic party, crowd without political status, and lots of academic consultants from foreign countries. The interaction between cultural institutions and Horqin shaman is positive,many well-known shamans are certified by inheritors, and the government does not interfere in their daily religious activities.
The contents of the footnote have been rewritten in the text.
Point.2
88-89 – I am confused – ordinary hospitals do not “handle” things such as traffic accidents? This needs to be clarified – what does the author really mean?
Response .2
There is a mistake, such as traffic accidents should be such as traffic accidents without harm of the body.
Point.3
146-47 – what happens to the animal after the sacrifice? Are the parts of the animal used?
Response.3
The animal will be as a sacred meal and eat by all the people present.
Point.4
308 – serve the masses – how is this reflected/related to “Serve the people” – wei renmin fuwu – does this aspect endear Shamanism a bit to the CCP?
Response .4
As many shamans often say, what they do is serve the people in need around them. In fact,a lot of people don't want to be shamans because if someone comes to you for help you have to help, you can't refuse, or you'll get some punishment from the shaman god Hettugen. These help-seekers take up shaman's time to earn money and support the family, leading to a poor economy for the shaman's family.
Many of the ideas of CCP come from the private sector and are not created by them out of thin air.
Round 2
Reviewer 2 Report
I have 2 suggestions based on the revised draft. The first is that I still don't see evidence of daily practice in cultivating xiuxing. The reader hears why it is important but, at present, the ethnographic data is missing. Where is the 'everyday' amongst the author's interlocutors?
Second, I recommend engaging with Uranchimeg Ujeed's work.